# Identification of predictive patient characteristics for assessing the probability of COVID-19 in-hospital mortality

**Bartek Rajwa**[1,2]*, **Md Mobasshir Arshed Naved**[3], **Mohammad Adibuzzaman**[4], **Ananth Y. Grama**[3], **Babar A. Khan**[5], **M. Murat Dundar**[6], **Jean-Christophe Rochet**[2,7]*

**1** Bindley Bioscience Center, Purdue University, West Lafayette, Indiana, United States of America, **2** Purdue Institute for Integrative Neuroscience, Purdue University, West Lafayette, Indiana, United States of America, **3** Dept. of Computer Science, Purdue University, West Lafayette, Indiana, United States of America, **4** Oregon Clinical and Translational Research Institute, Oregon Health and Science University, Portland, Oregon, United States of America, **5** Regenstrief Institute, Indianapolis, Indiana, United States of America, **6** Dept. of Computer and Information Science, IUPUI, Indianapolis, Indiana, United States of America, **7** Borch Dept. of Medicinal Chemistry and Molecular Pharmacology, Purdue University, West Lafayette, Indiana, United States of America

* brajwa@purdue.edu (BR); jrochet@purdue.edu (J-CR)

**Data Availability Statement:** The data used in this research has been provided through a data transfer

## Abstract

As the world emerges from the COVID-19 pandemic, there is an urgent need to understand patient factors that may be used to predict the occurrence of severe cases and patient mortality. Approximately 20% of SARS-CoV-2 infections lead to acute respiratory distress syndrome caused by the harmful actions of inflammatory mediators. Patients with severe COVID-19 are often afflicted with neurologic symptoms, and individuals with pre-existing neurodegenerative disease have an increased risk of severe COVID-19. Although collectively, these observations point to a bidirectional relationship between severe COVID-19 and neurologic disorders, little is known about the underlying mechanisms. Here, we analyzed the electronic health records of 471 patients with severe COVID-19 to identify clinical characteristics most predictive of mortality. Feature discovery was conducted by training a regularized logistic regression classifier that serves as a machine-learning model with an embedded feature selection capability. SHAP analysis using the trained classifier revealed that a small ensemble of readily observable clinical features, including characteristics associated with cognitive impairment, could predict in-hospital mortality with an accuracy greater than 0.85 (expressed as the area under the ROC curve of the classifier). These findings have important implications for the prioritization of clinical measures used to identify patients with COVID-19 (and, potentially, other forms of acute respiratory distress syndrome) having an elevated risk of death.

## Author summary

Gaining insight into the patient attributes that are indicative of poor outcomes resulting from infections caused by highly lethal viruses such as SARS-CoV-2 is of paramount importance for hospitals and healthcare systems in order to adequately anticipate and

agreement (DTA) between the Regenstrief Institute, 1101 West 10th Street, Indianapolis, IN 46202, and Purdue University, West Lafayette, IN 47907. The data has been de-identified in compliance with 45 C.F.R. § 164.51. The title to the data is solely owned and remains with the Regenstrief Institute, and it has been offered to Purdue University as a service to the research community. As per the agreement, Purdue researchers are obligated to restrict access to and processing of the data to authorized personnel or employees who are required to process the data to conduct their work in connection with the purposes outlined in the DTA. Third-party researchers or scientists who wish to access these data for their own research purposes are encouraged to directly contact the Regenstrief Institute via the https://www.regenstrief.org/data-request/ form. Such requests will be assessed in accordance with the Institute's data-sharing guidelines, which demonstrate their commitment to supporting wider scientific collaboration while maintaining rigorous standards of data privacy and security.

**Funding:** BR, BAK and JCR gratefully acknowledge funding from the COMMIT (COvid-19 unMet MedIcal needs and associated research exTension) Program of the Gilead Foundation (grant number IN-US-983-6060). The sponsor had no involvement in the study's design, data collection and analysis, decision to publish, or in the preparation of the manuscript. Information about the Gilead Foundation is available at https://www.gilead.com/purpose/gilead-foundation.

**Competing interests:** The authors have declared that no competing interests exist.

address forthcoming outbreaks. Approximately 20% of the infections in the initial phase of the COVID-19 pandemic resulted in severe respiratory complications triggered by inflammatory responses. Moreover, individuals with severe COVID-19, particularly those with pre-existing neurodegenerative conditions, who were at a heightened risk, frequently displayed neurologic symptoms. Although there is a documented association between severe COVID-19 and neurologic issues, the underlying causes remain elusive. In this study, we examined the electronic health records of 471 patients with severe COVID-19 to identify clinical indicators that could predict mortality. Through machine learning models, we discovered a series of clinical features, notably those associated with cognitive dysfunction, that accurately predict mortality (area under the ROC curve > 0.85). These results underscore the significance of specific clinical indicators in recognizing patients at increased risk of mortality from COVID-19, thus enabling more focused and effective healthcare strategies.

# 1 Introduction

## 1.1 COVID-19 and neurologic symptoms

Patients with severe COVID-19 warranting hospital admission present with a variety of symptoms that are carefully evaluated by admitting physicians. These patient evaluations have led to the identification of specific comorbidities linked to severe forms of COVID-19 or fatal outcomes [1–3], including neurodegenerative disorders and dementia [4–6]. Although SARS-CoV-2 infection can cause neurologic symptoms by directly affecting the central nervous system (CNS), this phenomenon has only been shown in a very small subset of patients [7]. Instead, neurologic symptoms are more likely to occur due to indirect effects involving the strong innate immune response and cytokine storm caused by SARS-CoV-2 infection.

A cytokine storm is a hyperinflammatory response to an infection caused by a sudden spike in levels of pro-inflammatory cytokines and chemokines, including IL-1, IL-2, IL-4, IL-6, IL-7, IL-8, IL9, IL-10, IL-18, granulocyte stimulating factor (G-CSF), IP-10, monocyte chemoattractant protein (MCP)-1, MCP-3, macrophage inflammatory protein 1 (MIP-1A), cutaneous T-cell attracting chemokine (CTACK), IFN-γ, and TNF-α. In turn, this phenomenon can result in overwhelming systemic inflammation, acute respiratory distress syndrome (ARDS), and multi-organ failure [8]. Evidence suggests that a cytokine storm can trigger various neurologic symptoms, ranging from headaches, dizziness, and disorientation to convulsions or seizures [8].

## 1.2 Approach to prediction and feature selection

In this report, we present the results of a study aimed at establishing a link between COVID-19 symptoms observed upon patient admission (within the first 24 h of hospitalization) and the risk of patient death. We were particularly interested in the predictive value of easily observable neuropsychiatric symptoms such as disorientation, cognitive impairment, and delirium. Our strategy involved a data-driven discovery of predictors. Instead of postulating *a priori* a feasible set of clinical features likely to be associated with mortality and then testing the resulting hypotheses using a standard generalized linear model (GLM) approach, we retained all possible clinical features for the analysis. We used the disease outcome to train a supervised classifier with feature ranking and selection capability. When an explainable classifier achieved high accuracy, we queried it to determine which feature combination was responsible for its

strong performance. The final output was a set of hypotheses (that could be addressed in future intervention studies) positing that the identified predictive combinations of clinical features were causally linked to mortality or that common latent factors influenced both the mortality as well as the detected clinical characteristics.

We utilized two independent approaches to quantify the predictive power of the observations collected at the time of patient admission, or within the first 24 h of hospitalization: elastic-net regularized logistic regression (LR-ENET) and XGBoost classification [9, 10]. The use of both methods was followed by an analysis of the relative contributions of the selected features ("explanatory variables") to the prediction via the SHapley Additive exPlanations (SHAP) approach. The SHAP factors provide a valuable tool for interpreting the significance of each clinical feature in a model. By assigning a value to each variable based on its average contribution to the outcome across all possible combinations, this method offers a comprehensive way to understand the relative importance of each feature [11–13]. This is particularly useful in complex scenarios where the features interact in intricate ways.

The described analyses led to the identification of several important predictive features, including neurologic symptoms.

### 1.3 Related research

Data-driven investigation of COVID-19 mortality employing machine learning tools (such as XGBoost) and feature explanation methods (SHAP values) has been demonstrated before for the processing of clinical laboratory results [14–16], for predicting death outcomes [17–19], demonstrating links between socioeconomic disparities and COVID-19 spread [20], and showing the impact of COVID-19 on mental health in self-identified Asian Indians in the USA [21]. Several other articles using SHAP were reviewed recently by Bottino et al. [22].

## 2 Methods and datasets

### 2.1 Ethics statement

This study, identified by IRB protocol #2004316653, was approved by the Indiana University Institutional Review Board. As the study procedures involved the use of a deidentified electronic medical record database, consent was not required.

### 2.2 Dataset description

Electronic health record (EHR) data were obtained from 471 patients with severe SARS-CoV-2 infection. The patients were admitted to the intensive care units (ICUs) of IU Health Methodist Hospital and Sidney & Lois Eskenazi Hospital, both in Indianapolis, Indiana, between March 2020, and August 2020. 399 patients were eventually discharged, whereas 72 patients died. The demographic characteristics of the cohort are shown in Table 1. 246 patients self-identified as *Black* or *African American*, and 196 patients identified as *white*. 245 of the patients were females, and 226 were males. There was no statistically significant difference in age between the African-American and white patients. However, patients who identified as Hispanic or Latino were significantly younger than other patients ($p < 0.001$).

### 2.3 Preprocessing

The original dataset consisted of data collected at multiple time points during the patients' treatment. We used only the data from the first time point (i.e., the initial evaluation and the subsequent measures up to 24 h), consisting of the earliest available diagnostic characteristics. We retained the minimal and maximal values if multiple measurements and/or laboratory

**Table 1. General demographic characteristics of the investigated cohort.**

| Race | Female | | | | | Male | | | | |
|---|---|---|---|---|---|---|---|---|---|---|
| | *Alive* | *Alive %* | *Died* | *Died %* | *Total* | *Alive* | *Alive %* | *Died* | *Died %* | *Total* |
| Asian | 3 | 100.0% | 0 | 0.0% | 3 | 5 | 83.3% | 1 | 16.7% | 6 |
| Black or Afr. Amer. | 123 | 87.9% | 17 | 12.1% | 140 | 85 | 80.2% | 21 | 19.8% | 106 |
| Refused to identify | 1 | 33.3% | 2 | 66.7% | 3 | 2 | 100.0% | 0 | 0.0% | 2 |
| Unknown | 10 | 100.0% | 0 | 0.0% | 10 | 5 | 100.0% | 0 | 0.0% | 5 |
| White | 77 | 86.5% | 12 | 13.5% | 89 | 88 | 82.2% | 19 | 17.8% | 107 |
| Total | 214 | 87.3% | 31 | 12.70% | 245 | 185 | 81.9% | 41 | 18.1% | 226 |

results were provided. Because many features describe the patient's status as very detailed and granular, some binary factors were observed for only a few (one or two) patients. These variance-deficient features were removed to prevent the model from outfitting, even though they might have been informative. Another step of feature engineering (not pursued in this study) is likely needed to construct virtual features that summarize these descriptors.

The presence of multiple correlated features related to delirium is an example of the described problem. We found that the large number of delirium-related descriptors would lead to the emergence of 29 distinct categories. However, because over 300 cases showed no evidence of delirium, an alternative approach would be to combine all of the positive categories into one (e.g., "some evidence of delirium"). In our LR-NET and XGBoost models, this engineered feature was not included. Nonetheless, a distinct analysis was carried out to assess the predictive value of the "delirium" secondary feature, with its findings detailed in the "Results" section.

## 2.4 Logistic regression with elastic-net regularization model

We established a set of relevant diagnostic features by implementing an *ante-hoc* explainable, predictable statistical model with embedded feature selection capability. We utilized utilize a logistic regression model regularized with a ridge ($\ell_2$), LASSO ($\ell_1$), or a combination of both penalties (elastic net) [9, 23, 24]. This approach allowed us to (1) create a simple model capturing all the significant sources of variability, incorporating all of the diverse clinical descriptors/features, and (2) perform simultaneous feature selection and feature ranking, allowing identification of the major drivers of correct prediction. It is worth highlighting that the inclusion of $\ell_1$ regularization may generate instability within the feature selection process [25]. However, this challenge can be addressed by re-executing the feature selection multiple times. Please refer to the S1 Appendix for details.

## 2.5 XGBoost model

We utilize the extreme gradient boosting decision tree (XGBoost). This method developed by Chen and Guestrin, is a highly effective, portable, and scalable machine learning system for tree boosting that is optimized under the Gradient Boosting framework [10]. It combines a series of low-accuracy weak classifiers using the gradient descent architecture to produce a strong classifier with higher classification performance.

## 2.6 SHapley Additive exPlanations (SHAP) values

The subsequent analysis of features identified by LR-ENET and XGBoost is performed using approach based on cooperative game theory concept known as SHAP (or Shapley) values. The

subsequent analysis of the features identified by LR-ENET and XGBoost was performed using an approach based on the cooperative game theory concept known as SHAP (or Shapley) values [11–13, 26]. The SHAP values can determine the importance of a feature and its directionality influence by comparing what a model predicts with and without that feature for each observation in the training data and calculating the marginal contribution [12].

In simple terms, the Shapley analysis is a method that allocates "credit" or "blame" fairly among a set of contributing factors or variables in a model. This is particularly useful in clinical research where numerous variables can influence an outcome, and it can be challenging to determine the importance or contribution of each variable, especially when they interact in complex ways.

The Shapley factor analysis addresses this challenge by considering all possible combinations of variables and their contributions to the accuracy of a model or multiple complementary models. Essentially, it answers the question: "What is the average contribution of each variable to the outcome when all possible combinations of variables are taken into account?" Therefore, each variable's SHAP can be interpreted as its average marginal contribution across all possible combinations of variables. Even with non-linear and complex interactions between features, SHAP values remain meaningful and interpretable.

## 3 Results

### 3.1 Patients characteristics

Patients who died of COVID-19 were significantly older than patients who survived ($p<0.001$) in both the female and male groups. The average Braden score was slightly lower among patients who died (females: $p=0.035$, males: $p=0.021$). There was no observable difference between the mean Glasgow Coma scores of patients who died or survived (females $p=0.14$, males: $p=0.15$). Patients who presented with clear speech and, therefore, presumably had intact cognitive ability were significantly over-represented among those who survived (odds ratio of being admitted with clear speech for surviving patients: 5.65 ($p<0.001$) for females, 3.6 ($p<0.001$) for males). See Table 2 for a summary of the findings.

Delirium-related symptoms were noted across all patient age groups, but were particularly prevalent among the elderly. Interestingly, when the cohort was stratified by gender, older female patients were found to be at a significantly higher risk of developing delirium, whereas no statistically significant increase in risk was observed among their male counterparts. See Table 3 and Table A in the S1 Appendix. When adjusting for age, there was no difference in delirium occurrence between patients in the *white* and *African American* subcohorts ($p=0.52$).

The composite feature that describes the presence of any delirium-related symptoms has been identified as an excellent univariate predictor of patient outcomes. Patients presenting delirium symptoms have a significantly higher probability of death, a relationship that has been observed in both female and male cohorts. See Table 4.

To compare the formulated feature selection models to a benchmark, we created a simple GLM (non-regularized regression) employing the composite delirium feature, sex, race,

**Table 2. Mean ages, Braden scores, Glasgow coma scores, and probabilities of demonstrating a clear speech pattern for patients in the investigated cohort.**

| Sex | Status | Age, mean | Age, IQR | Braden score, mean | Coma score, mean | $\mathcal{P}$(Clear speech) | (95% CI) |
|---|---|---|---|---|---|---|---|
| Female | alive | 55.08 | 25.61 | 16.92 | 13.80 | 0.84 | (0.79, 0.88) |
| | died | 80.27 | 19.35 | 13.90 | 12.84 | 0.48 | (0.32, 0.65) |
| Male | alive | 56.70 | 20.18 | 17.11 | 14.14 | 0.82 | (0.76, 0.87) |
| | died | 75.31 | 12.85 | 14.15 | 13.29 | 0.56 | (0.41, 0.7) |

**Table 3. Occurrence of delirium in different age groups of male and female patients in the study.**

| Age | Female | | | | | Male | | | | |
|---|---|---|---|---|---|---|---|---|---|---|
| | *No delirium* | *No delirium %* | *Delirium* | *Delirium %* | *Total* | *No delirium* | *No delirium %* | *Delirium* | *Delirium %* | *Total* |
| [17.1, 51.8) | 84 | 92.30% | 7 | 7.70% | 91 | 59 | 89.40% | 7 | 10.60% | 66 |
| [51.8, 66.9) | 66 | 91.70% | 6 | 8.30% | 72 | 72 | 84.70% | 13 | 15.30% | 85 |
| [66.9,101.5] | 57 | 69.50% | 25 | 30.50% | 82 | 60 | 80.00% | 15 | 20.00% | 75 |
| Total | 207 | 84.50% | 38 | 15.50% | 245 | 191 | 84.50% | 35 | 15.50% | 226 |

**Table 4. Probability of death given signs of delirium at the hospital admittance and accompanying lower (LCL) and upper (UCL) 95-percentile confidence limits.**

| Sex | Status | $\mathcal{P}$(Death) | LCL | UCL |
|---|---|---|---|---|
| Female | No delirium | 0.08 | 0.05 | 0.12 |
| | Delirium | 0.39 | 0.25 | 0.56 |
| Male | No delirium | 0.13 | 0.09 | 0.19 |
| | Delirium | 0.46 | 0.30 | 0.62 |

Braden score, and age category discretized into three tertiles: *young* [<55.7], *middle* [51.7, 66.8] and *older* [>66.9]. The mathematical specification of the model is provided in the S1 Appendix (Equation G).

Inspection of the model demonstrates that age is the most important predictive factor, followed by delirium symptoms and the Braden score. There is no significant difference between patients of different races. Table B in the S1 Appendix shows the detailed results of the statistical analysis, and Table C presents the average marginal effects (AME) computed from the model. After fitting, the model has an AUC of 0.87, a sensitivity of 0.94, and a specificity of 0.3. Post-hoc analysis demonstrated that after controlling for race, sex, age group, and the Braden score, the odds of death for patients exhibiting delirium symptoms increased by 5.23 ($p<0.001$).

### 3.2 Regularized logistic regression model results

LR-ENET model training was performed using a grid search through the space of parameters $\lambda$ and $\alpha$. An example of a training grid is illustrated in Fig 1.

The performance of the trained LR models is demonstrated in Fig 2. The classifier performed well and was consistently able to reach an AUC of approximately 0.9.

Each regularized logistic regression model in the ensemble was trained with the objective of limiting the feature count to around twenty. To address data imbalance, the training and feature selection process included an augmentation step where synthetic data points were generated. This was achieved by applying the ROSE algorithm [27] directly to each set of bootstrapped data during the 0.632 bootstrap cross-validation training. This approach not only addressed the imbalance in the data but also introduced additional variability, presenting a robust challenge to the classifier.

After each run, absolute values of the different LR-ENET models were collected and scaled to [0, 1] intervals. These measures were considered reflective of the predictors' importance. Due to instability, several lower-performing predictors were observed with non-zero coefficients, though only in a few runs. On the other hand, consistent predictors emerged in most or all of the runs. All of the scaled predictive importance values were subsequently analyzed, and the top 20 were picked for the final selection.

The results illustrating the identified predictive EHR features are summarized in Fig 3. As mentioned previously, due to the inherent instability of sparse classifiers, each run may result

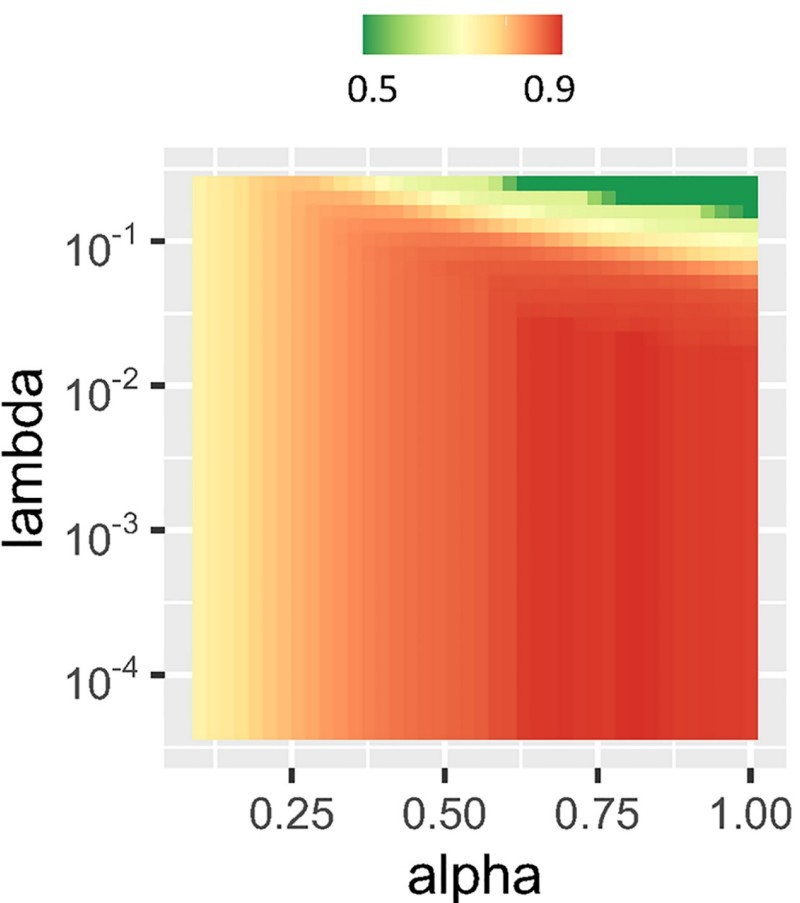

**Fig 1. Changes in the performance of the elastic-net regularized model in the classification of COVID-19 patients, expressed as AUC, given different values of α and λ parameters.**

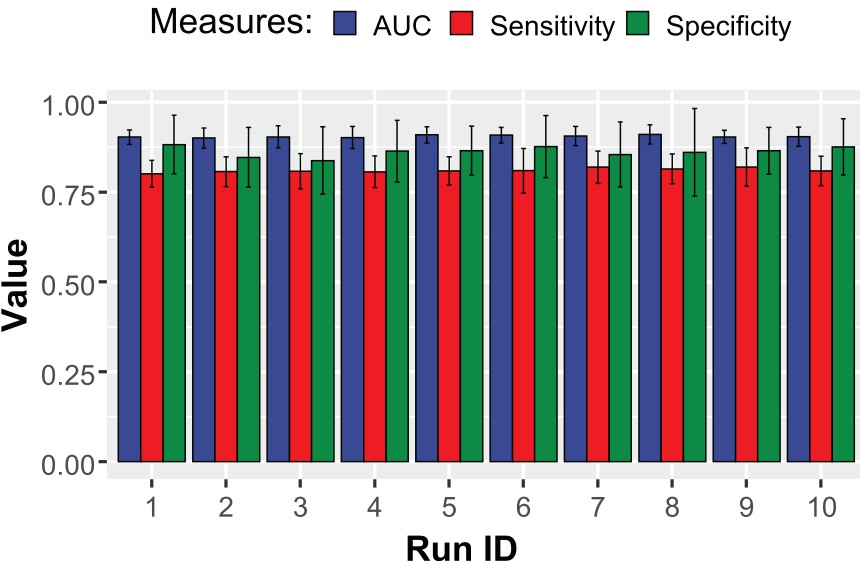

**Fig 2. Performance of the elastic-net regularized regression model in the classification of COVID-19 patients.** Ten independent rounds of model training are shown.

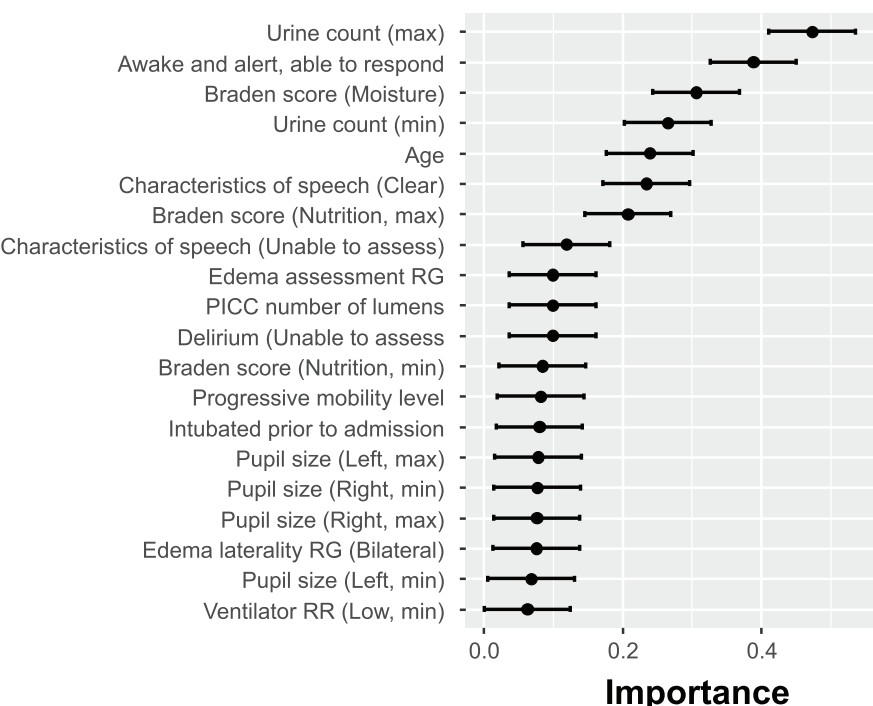

**Fig 3. Importance of elastic-net selected features shown as the normalized absolute value of the regression coefficients.** Multiple runs of the feature selection were executed to alleviate the inherent instability of the sparse models.

in a slightly different order of selected features. This is expected since informative feature sets may have multiple subsets that lead to the same classification success. Therefore, formally, the SHAP analysis should be performed for every separate feature selection run. However, to obtain some preliminary insight into the explanatory power of the selected features, we conducted the SHAP analysis for only one of the ten LR-ENET-trained classifiers, which well represented the mean feature importance. The results are demonstrated in Fig 4.

### 3.3 XGBoost model results

XGBoost performed comparably to the LR-ENET algorithm, although it is noted that the acquired specificities were often lower. We optimized the XGBoost hyperparameters to maximize the AUC as opposed to explicitly minimizing the number of utilized features. Therefore, the algorithm was allowed to employ as few or as many features as were required to optimize its performance.

The XGBoost model can be used as a feature selection wrapper. In the process of training the features are selected in ignored in the created trees. On the basis of that, the XGBoost method ranks the most significant characteristics according to "Gain," "Cover," and "Frequency." The gain reflects how crucial a characteristic is for making a branch of a decision tree pure. Coverage measures the proportion of observations affected by a feature. A feature's frequency is the number of times it is used in all created trees (See Fig 5).

The XGBoost algorithm was not directly constrained by the number of used features. The performance of the XGBoost classifier is shown in Fig 6. The XGBoost-discovered features were exposed to the SHAP analysis, the results of which are demonstrated in Fig 7.

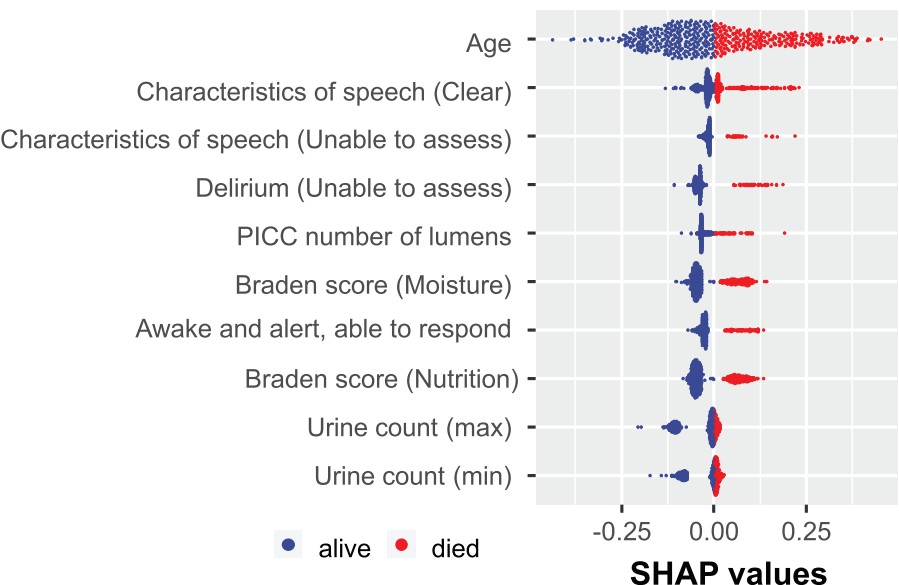

**Fig 4. Example of the SHAP value distribution for all of the tested patients and the augmented dataset.** The visualization was created based on one of the trained LR-ENET classifiers.

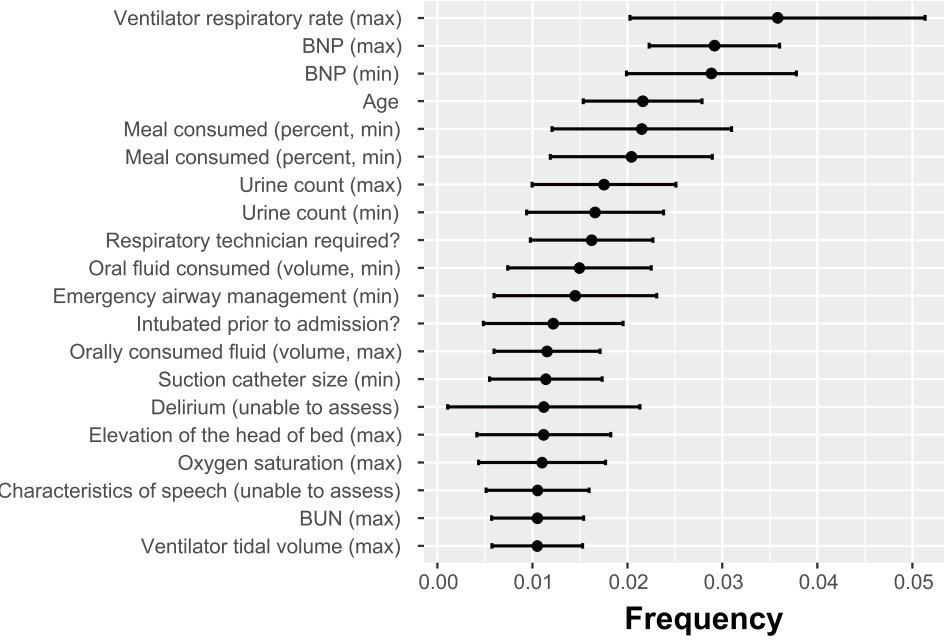

**Fig 5. Importance of features employed by XGBoost algorithm expressed as the frequency at which each feature was used in the created classification trees.**

### 3.4 Discovered predictive features

Both feature-selection strategies identified a number of clinical characteristics that reflect overall patient health, cognitive status, and hospitalization risks associated with the patient's condition. Here, we list some of the identified features, with a particular focus on those discovered

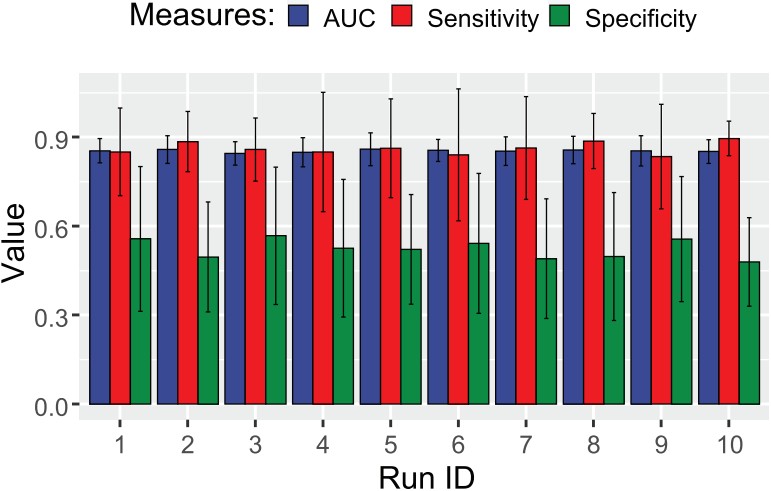

**Fig 6. Performance of the XGBoost model in repeated independent cross-validation rounds classifying the COVID-19 patients.**

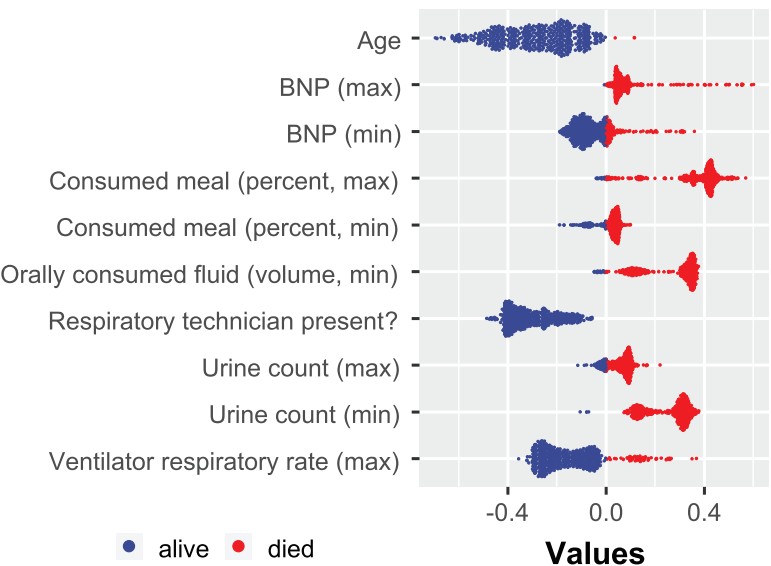

**Fig 7. Example of SHAP value distribution for all of the tested patients.** The visualization was created based on one of the trained XGBoost classifiers.

by the LR-ENET model, which demonstrated superior AUC, sensitivity, and specificity compared to the XGBoost approach.

**CNS and cognition-related features.** A set of features related to the patient's neurologic status was selected. The LR-ENET approach generated a model richer in these features than the XGBoost model.

- **Inability to assess the patient's speech** may be caused by neurologic symptoms (such as loss of consciousness or the presence of delirium), but it may also be related to intubation. This

EHR observation represents a surrogate measurement of the patient state in both instances, and its existence significantly suggests an elevated mortality risk.

- **Unclear and slurred speech** observed by nurses or physicians could be a key indicator of delirium [28]. Delirium is related to adverse outcomes during hospitalization (e.g., increased risk of complications) in post-acute care settings, and long-term follow-up (e.g., prolonged cognitive and functional impairment). The lack of speech clarity is only a surrogate measure for delirium that might be considered low value given that the presence of delirium is another explicitly defined feature in the analyzed dataset. However, it should be noted that there is a significant rate of under-recognition and lack of documentation of delirium in admitted patients, with less than 3% of cases documented by International Classification of Diseases-9 (ICD-9) codes in patients' medical records [28, 29]

- **Any evidence of delirium.** Among the delirium-related characteristics included in the dataset is the expressly stated "Presence of delirium." However, the feature selected as predictive was "Inability to assess delirium," which may appear odd and counterintuitive. Examining the explicit delirium feature reveals a biserial correlation between the feature and the patient's registered death of only 0.13, whereas the inability to assess delirium scores -0.25. This curious selection may be partially explained by the procedure required to assess delirium, which also explains the under-recognition of delirium in general.
The Confusion Evaluation Method, the most used delirium assessment tool, requires an in-person, bedside discussion with the patient [30]. Because delirium fluctuates, interview-based approaches may overlook delirium that occurs beyond bedside interviews. It has been reported that manual searches of all records (e.g., nursing and physician notes, discharge summaries) may allow the determination of signs of delirium despite the lack of explicit notes in the records [28]

- **"Awake and able to response"** is another important neurologic assessment feature. A patient is scored positively if considered "awake," "able to respond" (i.e., responding appropriately), and "oriented" (aware of self, place, and time) [31]. However, the patient's situation may change on the first day after admission. Therefore, patients who were sedated and unable to be subjected to the neurologic examination may demonstrate full-strength cognitive abilities later. Thus, yet again, this feature provides value in combination with other features and cannot be considered in isolation. Interestingly, the Glasgow coma scale (GCS), which a measure used to determine the level of consciousness in trauma or critically ill individuals with impaired consciousness and which was also available in the dataset, has not been utilized by the models.

**Patient frailty-related features.** Several identified features described the patient's overall condition and frailty upon hospital admission.

- **Braden score** describes the frailty of the patients. The Braden scale was created to identify early pressure, sore-prone patients. Six sub-scales of the score measure sensory perception, skin wetness, activity, mobility, friction and shear, and nutrition [32]. Although ample evidence exists for the usefulness and applicability of the Braden scale in predicting patients' conditions during hospitalization [33], the scale has been criticized for lacking explainability and detail from the machine learning perspective [34]. On the other hand, a retrospective study of 146 COVID-19 patients demonstrated that the Braden score is indeed helpful for risk stratification at hospital admission, as the mortality among patients with BS≤15 was significantly higher than in patients with BS>15 [35].

In our research context, Braden score features also appear to identify particularly vulnerable patients. Interestingly, only two sub-scales (moisture and nutrition) have been included by the LR-ENET classifier in the final model. On the other hand, the XGBoost model did not rely on Braden scores.

- **Number of lumens** Multiple studies have demonstrated a substantial correlation between the number of PICC lumens and the risk of complications, including central-line associated bloodstream infection (CLABSI), venous thrombosis, and catheter occlusion [36, 37].

- **The requirement for a respiratory technician** to be present during the transportation of the patient is yet another predictive EHR feature that communicates the severity of the patient's condition.

- **The urine voiding count feature** is connected with the lower urinary tract symptoms. The medical literature describes an association between LUTS and COVID [38–40]. According to these reports, there was a high prevalence of abnormal urinary storage symptoms, urine frequency, urgency, and urinary incontinence among the SARS-CoV-2-infected patients. The data indicate that the majority of COVID-19 patients may experience increased urination frequency, nocturia, and urgency during the infection. Also, patients with urine storage symptoms were found to have considerably higher COVID-19 severity levels than those without urine storage symptoms [40]. The urinary symptoms might be caused directly by inflammation or indirectly by COVID-19-related general dysfunction in the autonomic nervous system [41].

   **Clinical laboratory test results.**   Report on the patient's physiological status. Specifically, two test results were identified by our feature selection methods.

- **The measurement of urea nitrogen in serum or plasma** (BUN SerPl test) assesses the kidney's function. High urea nitrogen levels in the BUN test indicate problems with renal function or a reduction in blood supply to the kidneys. A reduction in urea nitrogen, as measured by the BUN test, indicates serious liver illness or malnutrition. The outcomes of the BUN test were put into the XGBoost model's identified collection of features. However, they were not among the top 20 features identified by LR-ENET. In the analyzed cohort, there was a significant difference in test findings between patients who died and those who survived ($p<0.001$). These observations are consistent with literature reports [42, 43].

- **Brain natriuretic peptide (BNP)** is an active fragment (1–32) of the cardiac cell-produced ProBNP. It is elevated in right-sided and left-sided heart failure, as well as systolic and diastolic heart failure. It is, therefore, used to identify and treat heart failure. The BNP test was recognized as an important feature by the XGBoost-feature selection, but not the LR-ENET. In the tested cohort, the values of the BNP test were substantially higher for male patients who died ($p<0.001$) but not for females ($p=0.23$). Others have recently postulated that BNP should be considered an early predictor of clinical severity in patients with COVID-19 pneumonia [44].

## 4 Discussion

### 4.1 Clinical feature selection

When analyzing disease mortality causes, the search for predictive factors typically begins with the formulation of a hypothesis based on domain knowledge of the underlying diseases and initial preliminary evidence, such as case studies and anecdotal reports. This hypothesis-driven process is philosophically well-established and operationally widely accepted. Despite the fact

that this conventional path of hypothesis-driven research has been challenged numerous times in recent years, particularly by the rise of genomics, for many researchers it is virtually synonymous with the scientific method itself [45]. The alternative paradigm, often referred to as data-driven research, begins with an agnostic stance that does not involve a preconceived hypothesis and instead employs either a data reduction process that results in the emergence of a model or a supervised model-building process that reveals predictive features that explain observed outcomes.

Here, we examined three methods for identifying factors predictive of mortality among COVID-19 patients with severe disease necessitating hospitalization. As a baseline, we utilized a standard statistical technique for formulating a hypothesis and generating or selecting hypothesized predictive factors through a non-regularized GLM framework. In carrying out this method, we used a composite delirium factor, which is a combination of multiple EHR characteristics/symptoms associated with delirium occurrence. We also accounted for age, race, and sex, which we recognized as notable confounding variables plausibly related to the outcome. The developed model revealed a significant increase in the likelihood of death for hospitalized patients exhibiting any symptom of delirium.

Subsequently, we developed two feature discovery models using two well-established machine learning techniques. The first model utilized regularized elastic-net logistic regression. A sparse collection of predictors was generated using the model's inbuilt feature selection capability. However, as previously established, the sparse models are inherently unstable, and the selection of reliably predictive features necessitated many model runs and a compilation of the findings [25]. The most consistently predicted variables across numerous runs were evaluated using the SHAP method to obtain insight into their local significance for patient classification.

The second machine learning model utilized the XGBoost tree learning technique. Unlike the LR-ENET approach which enforced sparsity, this method freely utilized all available clinical features. After multiple rounds of independent training and cross-validation, the algorithm demonstrated high stability, ultimately producing very similar results. To represent feature importance, data from the internals of trained classifiers (gain and frequency) were extracted. A secondary XGBoost model was then trained using the top features, and its SHAP values were assessed.

Our findings demonstrate that the two distinct classifiers relied on very different sets of predictive features during optimization and training. Age, surrogate measures for the patients' cognitive status (neurologic observations), features broadly describing the patients' overall condition upon admission (such as the Braden score), and features associated with a risk of serious complications requiring hospitalization (such as the number of catheter lumens) were the descriptors that the LR-ENET model selected. It is interesting to note that common clinical laboratory test results were not chosen during the feature selection process by LR-ENET.

XGBoost, on the other hand, placed considerable emphasis on laboratory test results, including values obtained from the BUN and BNP tests. XGBoost captured the patient's overall condition by analyzing variables such as oral fluid consumption, oxygen saturation, ventilator use, etc. Despite the fact that the features chosen by XGBoost are frequently more quantitative and objective, the overall performance of the XGBoost model was marginally inferior in terms of specificity (such as in the case of the laboratory test results).

## 4.2 Limitations

It is important to acknowledge that the study presented here has certain critical limitations that require a thorough understanding.

The study lacks consideration of the patients' comorbidity status, which is a complex and multifaceted area. It is widely acknowledged that patients with severe and potentially fatal cases of SARS-CoV-2 infection typically have multiple comorbidities [46–52]. This is also true for our specific cohort, and has been demonstrated by the data continuously collected by the Regenstrief Institute on patients from Indiana [53, 54]. However, due to the limited sample size and diverse range of comorbidities present, we would not be able to draw any conclusive inferences about which specific comorbidities may impact the likelihood of mortality.

The other limitation stems directly from the question of the extent to which the deaths among the patients in the study cohort can be directly associated with COVID-19, rather than other unrelated factors. The patients were admitted to the hospital's ICU due to the onset of symptoms, not because of results of a screening test. As the admission occurred before the availability of rapid testing, it was assumed that the severity of symptoms was at least correlated with their later confirmed COVID status. It is important to bear in mind that in the early phase of the pandemic, the admission of patients without severe respiratory distress symptoms was minimized across hospital systems in the US and elsewhere [55, 56]. It would be impossible to ascertain whether every individual patient in the cohort who died indeed died "from" COVID or rather "with" COVID unless an autopsy followed by a detailed inquest were performed in every case. However, the studies on overall mortality in the US indicate an underestimation, rather than an overestimation, of COVID mortality [57, 58].

Fillmore et al. showed that the number of hospitalized patients testing positive for COVID-19 without exhibiting related symptoms rose in tandem with greater virus exposure and advancements in vaccination programs [59]. Consequently, it is now more crucial to differentiate between patients who died "with" COVID-19 and those who died "due to" COVID-19, especially when analyzing recent data, as opposed to the historical data from the early and mid-2020s, which we used.

The described limitation is related to the wider issue of clinical coding, as possible errors in coding for mortality and morbidity significantly impact any use of machine learning in clinical decision support. We recognize that potential inaccuracies in ICD code assignment, and resulting data inconsistencies, may lead to flawed machine learning models, yielding unreliable predictions.

Lastly, it is important to note that our research utilizes a specifically narrow understanding of causal links. While feature importance scores as well as Shapley factors are valuable tools for attributing potentially causal effects to individual variables or factors within a presupposed causal model—particularly in cases where multiple variables are involved—we do not claim that a high Shapley value (or another metric of statistical association) is, on its own, proof of a causal relationship [60].

However, the measures of feature importance can indicate to which extent changing the variable (intervention) may contribute to changes in the outcome. This aligns with the causal inference goal of understanding the effect of interventions. They also help to decompose these interactions between variables. This is crucial in causal inference, as it aids in understanding how combined interventions might affect an outcome.

The measures of clinical feature importance do inspire counterfactual reasoning, as they allow us to consider the contributions of variables under different intervention scenarios. For instance, the significant impact of delirium suggests that preventing its onset could be a compelling strategy for intervention. Such variables can then be further investigated for causal relationships using a randomized clinical trial setting.

In essence, the clinical features (surrogate measures) identified as important indicators for certain underlying conditions, as detailed in the "Results" section, highlight noteworthy statistical associations. It's important to note that the presence of these associations alone does not

establish causality. For that, interventions are necessary, and our model offers recommended intervention strategies that show the greatest potential.

## 5 Conclusions

Risk stratification of hospitalized COVID-19 patients is crucial for informing individual treatment decisions that also account for resource allocation. Multiple risk models proposed so far have been based on mechanistic hypotheses regarding the SARS-CoV-2 mode of action, the association between comorbidities and observed outcomes, as well as hypothesized models of disease progression. In this report, we demonstrated the use of a machine learning-based, hypothesis-agnostic methodology for the discovery of predictive risk factors, which produces an easily understandable, observable, explainable, and actionable set of clinical features that may cause or be closely associated with in-hospital COVID-19 mortality.

This work sets the stage for future intervention-based studies aimed at addressing hypotheses that the clinical features identified here are causally linked to COVID-19 mortality. Even in the absence of established causal relationships, our findings suggest that analyses of these features should be prioritized to identify patients with COVID-19 (and, potentially, other forms of acute respiratory distress syndrome) having an elevated risk of mortality.

## Supporting information

**S1 Appendix. Table A.** Summary of differences in the occurrence of delirium among the three age groups of patients. **Table B.** Statistical summary of the regression model shown in S1 Appendix. Equation G. **Table C.** Average marginal effect (AME), *p*-values, and confidence intervals associated with the factors incorporated in the benchmark model introduced in S1 Appendix. Equation G. The AME represents the average effect of a unit change in a predictor variable on the predicted outcome, averaged across all observations in the dataset. It provides a straightforward measure of the influence of each independent variable, revealing the expected average change in the dependent variable with a one-unit increase in that independent variable, while keeping all other variables constant.
(PDF)

## Author Contributions

**Conceptualization:** Bartek Rajwa, Babar A. Khan, M. Murat Dundar, Jean-Christophe Rochet.

**Data curation:** Bartek Rajwa, Md Mobasshir Arshed Naved, Mohammad Adibuzzaman, M. Murat Dundar.

**Formal analysis:** Bartek Rajwa, Md Mobasshir Arshed Naved, M. Murat Dundar.

**Funding acquisition:** Bartek Rajwa, Babar A. Khan, M. Murat Dundar, Jean-Christophe Rochet.

**Investigation:** Bartek Rajwa, Mohammad Adibuzzaman, Babar A. Khan, M. Murat Dundar, Jean-Christophe Rochet.

**Methodology:** Bartek Rajwa, Md Mobasshir Arshed Naved, Mohammad Adibuzzaman, Ananth Y. Grama, Babar A. Khan, M. Murat Dundar, Jean-Christophe Rochet.

**Project administration:** Babar A. Khan, Jean-Christophe Rochet.

**Resources:** Ananth Y. Grama, Babar A. Khan.

**Software:** Bartek Rajwa, Md Mobasshir Arshed Naved, Ananth Y. Grama.

**Supervision:** Bartek Rajwa, Jean-Christophe Rochet.

**Visualization:** Bartek Rajwa.

**Writing – original draft:** Bartek Rajwa, Md Mobasshir Arshed Naved, Babar A. Khan, Jean-Christophe Rochet.

**Writing – review & editing:** Bartek Rajwa, Jean-Christophe Rochet.

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
