## [Decision Letter · Decision Letter 0]

16 Oct 2023

PDIG-D-23-00264

Identification of predictive patient characteristics for assessing the probability of COVID-19 in-hospital mortality

PLOS Digital Health

Dear Dr. Rajwa,

Thank you for submitting your manuscript to PLOS Digital Health. After careful consideration, we feel that it has merit but does not fully meet PLOS Digital Health's publication criteria as it currently stands. Therefore, we invite you to submit a revised version of the manuscript that addresses the points raised during the review process.

Please submit your revised manuscript within 60 days Dec 15 2023 11:59PM. If you will need more time than this to complete your revisions, please reply to this message or contact the journal office at digitalhealth@plos.org. Please include the following items when submitting your revised manuscript:

We look forward to receiving your revised manuscript.

Kind regards,

Gilles Guillot

Academic Editor

PLOS Digital Health

Journal Requirements:

1. We ask that a manuscript source file is provided at Revision. Please upload your manuscript file as a .doc, .docx, .rtf or .tex.

Additional Editor Comments (if provided):

The manuscript has been evaluated by two reviewers and they both have substantial concerns about it. 

It is an important topic, the ms contains a number of interesting ideas and the reviewers do not close the door to a resubmission. 

For these reasons, I would suggest to the authors to assess carefully whether they are able to adess reviewers' concerns,and if so, to submit a revision.

Specific comments:

In the overall clear and well written but too technical for the audience. 

Equations would best appear only in an appendix. Technical terms should be defined/explained briefly. 

Section 2.5 contains a lot of ideas but is is too technical, difficult to follow and hard to relate to the problem at hand. 

I appreciate the effort made in producing high quality figures. They could be better leveraged with enhanced captions and comments in the main text. 

p.6 table 5: Please define Average marginal mean and how it is computed (btw, isn't it Average Marginal Effect as suggested by the seconf column name?)

There are two logistic regression (LR) models here (sections 2.3 and 3.1) and it is not always clear which model it is referred to later in the text. Also the wording GLM is sometime used instead of LR. Please chose one wording and stick to it consitantly. 

p.6 fig 1 the caption refers to \\gamma while it should presumably be \\lambda

p.7 what is the "LS algorithm"?

Reviewers' comments:

Reviewer's Responses to Questions

**Comments to the Author**

1. Does this manuscript meet PLOS Digital Health’s publication criteria? Is the manuscript technically sound, and do the data support the conclusions? The manuscript must describe methodologically and ethically rigorous research with conclusions that are appropriately drawn based on the data presented.

Reviewer #1: Yes

Reviewer #2: No

2. Has the statistical analysis been performed appropriately and rigorously?

Reviewer #1: Yes

Reviewer #2: No

3. Have the authors made all data underlying the findings in their manuscript fully available (please refer to the Data Availability Statement at the start of the manuscript PDF file)?

Reviewer #1: Yes

Reviewer #2: No

4. Is the manuscript presented in an intelligible fashion and written in standard English?

Reviewer #1: Yes

Reviewer #2: No

5. Review Comments to the Author

Reviewer #1: The authors suggest a predictive model for mortality due to COVID-19 among severely ill patients. While the topic is important, several questions arise that should be addressed: 

- The part on "Dataset description" should be in Results section. The Methods section should include a description of (i) how the data was acquired (ii) what were the inclusion and exclusion criteria and (iii) what variables were collected, including a clear description of the outcome variables.

- Patient characteristics are described in the results when compared between survivors and non-survivors. This is already a univariate analysis for testing association with mortality. However patient characteristics should be compared between the categories of the main independent variable, which is delirium. Such analysis is typically useful for indicating confounders and other effects to be accounted for in the multivariate analysis.

- As to the findings regarding delirium – could they be just another representation of an older age? Although age was included in the multivariate analysis, were there cases at all among the younger patients? A possible answer could be obtained by an additional subgroup analysis, assessing the effect of delirium among the older patients group only.

- Known risk factors for COVID-19 mortality were not accounted for. Were all patients comorbidity free? 

- "Importantly, Shapley values may have causal interpretations where the conventional “conditioning by observation” as in Pearl’s do-calculus, can be replaced by “conditioning by intervention” – please explain further – which of the effects is hypothesized as causal and why is it suitable to such definition. Was the causal effect estimated? If so please specify it clearly in the Results Section.

Reviewer #2: The manuscript under consideration presents results from developing classifiers for COVID-19 mortality amongst severe ICU COVID-19 cases using machine learning techniques. While the research topic is relevant, the manuscript has significant methodological, and reporting issues that prevent its current acceptance.

The methods section provides extensive detail on household methods, which might be better suited for supplementary material. Conversely, critical information necessary for evaluation is missing, making it impossible to interpret the results, e.g. information on

* Features definitions, engineering/grouping, what is available for selection, the coding system, what dataset is used and what is available in it (e.g. HES in the UK) – affects interpretation as a specific condition may have multiple distinct coding methods, diminishing their significance when examined separately.

* Given the relatively small sample size (N=471 with 72 cases), the XGB model may risk overfitting and limited generalisability.

* Defining COVID-19 mortality / clarifying if patients were admitted for COVID or other reasons, e.g. fatally-ingested toxic substances, but when admitted found to have COVID?

* Data collection time period – needed to account for potential variations in outcomes due to evolving treatment strategies and COVID variants.

* Handling of missing data

* Why this benchmarking model/particular features?

* Why is 20 predictors set as the limit for elastic-net?

* Is oversampling (ROSE) done before creating bootstrap samples? This can introduce bias and artificially inflate the AUC, as the OOB samples overlap with the training sample. 

The manuscript's claims that predictive combinations of clinical features are causally linked to mortality or that the study produces easily understandable and actionable clinical features lack sufficient support based on the presented analysis. As the authors mentioned, the different approaches selected “very different sets of predictive features" and that the selected features vary considerably depending on the seed. The selected features may also well be a proxy for other conditions. 

On top of the points above, the reporting lacks clarity and pertinent details, making it challenging to follow and evaluate the results.

In conclusion, I recommend rejecting the manuscript in its current form. Significant revisions to the structure, methodology, and reporting are necessary for reconsideration.

6. PLOS authors have the option to publish the peer review history of their article (what does this mean?). If published, this will include your full peer review and any attached files.

**Do you want your identity to be public for this peer review?** For information about this choice, including consent withdrawal, please see our Privacy Policy.

Reviewer #1: No

Reviewer #2: No

---

## [Decision Letter · Decision Letter 1]

6 Mar 2024

Identification of predictive patient characteristics for assessing the probability of COVID-19 in-hospital mortality

PDIG-D-23-00264R1

Dear Dr Rajwa,

We are pleased to inform you that your manuscript 'Identification of predictive patient characteristics for assessing the probability of COVID-19 in-hospital mortality' has been provisionally accepted for publication in PLOS Digital Health.

Best regards,

Gilles Guillot

Academic Editor

PLOS Digital Health

Reviewer Comments (if any, and for reference):

Reviewer's Responses to Questions

**Comments to the Author**

1. If the authors have adequately addressed your comments raised in a previous round of review and you feel that this manuscript is now acceptable for publication, you may indicate that here to bypass the “Comments to the Author” section, enter your conflict of interest statement in the “Confidential to Editor” section, and submit your "Accept" recommendation.

Reviewer #1: All comments have been addressed

2. Does this manuscript meet PLOS Digital Health’s publication criteria? Is the manuscript technically sound, and do the data support the conclusions? The manuscript must describe methodologically and ethically rigorous research with conclusions that are appropriately drawn based on the data presented.

Reviewer #1: Yes

3. Has the statistical analysis been performed appropriately and rigorously?

Reviewer #1: Yes

4. Have the authors made all data underlying the findings in their manuscript fully available (please refer to the Data Availability Statement at the start of the manuscript PDF file)?

Reviewer #1: Yes

5. Is the manuscript presented in an intelligible fashion and written in standard English?

Reviewer #1: Yes

6. Review Comments to the Author

Reviewer #1: (No Response)

7. PLOS authors have the option to publish the peer review history of their article (what does this mean?). If published, this will include your full peer review and any attached files.

**Do you want your identity to be public for this peer review?** For information about this choice, including consent withdrawal, please see our Privacy Policy.

Reviewer #1: No
